# A Study on the Preliminary Plan for Environmental Health in Seoul, Korea

**DOI:** 10.3390/ijerph192416611

**Published:** 2022-12-10

**Authors:** Jong-Seok Won, Hyomi Kim, Sang-Gyoon Kim

**Affiliations:** The Seoul Institute, 57 Nambusunhwan-ro, 340-gil, Seocho-gu, Seoul 06756, Republic of Korea

**Keywords:** Seoul Metropolitan Government, local plan for environmental health, environmental hazards, environmental health conditions, Seoul, basic policy direction, vision, goals, strategies, tasks

## Abstract

In the Republic of Korea, the Environmental Health Act was amended in January 2021, making it mandatory for each local government to establish a plan for environmental health. Accordingly, the Seoul Metropolitan Government (SMG) must establish the Local Plan for Environmental Health (LPEH) to protect citizens’ health from environmental hazards. The plan would support existing environmental health policies in Seoul to improve population health and achieve their goals. As a proof-of-concept to establish the LPEH, we developed a preliminary plan for environmental health in Seoul. We analyzed environmental health conditions of Seoul, identified driving conditions for execution of environmental health policies, set basic policy directions, and identified tasks needed to establish the preliminary plan. As a result, we established the vision and the goals of the preliminary plan. The vision is “a safe Seoul environment, healthy citizens”. The strategies are “active monitoring of environmental health issues”, “minimization of health damage and meticulous and systematic response”, and “building a foundation for environmental health”. To achieve the vision and the goals, we developed three strategies, eight tasks, and 25 sub-tasks. Under the preliminary plan we developed, we expect that SMG is able to protect citizens’ health from threats of environmental hazards; improve environmental health conditions, especially in susceptible populations such as infants; and promote environmental justice.

## 1. Introduction

The National Environmental Health Association defines Environmental Health as “the science and practice of preventing human injury and illness and promoting well-being by: identifying and evaluating environmental sources and hazardous agents and limiting exposures to hazardous physical, chemical, and biological agents in air, water, soil, food, and other environmental media or settings that may adversely affect human health” [1]. From this point of view, so far, there have been rare cases of large-scale environmental health problems such as LA smog or London smog threatening population health in Seoul. Nevertheless, the need for environmental health policies is emerging because daily exposure to environmental hazards may pose a threat to human health [2].

Although large-scale point sources such as industrial complexes and abandoned mines are not common in Seoul, mobile emission sources appear as major air pollution sources due to high traffic volume. Seoul has the highest population density among metropolitan governments in the Republic of Korea. To reduce environmental health threats due to air pollution, the Seoul Metropolitan Government (SMG) introduced various regulations to reduce air pollutant emissions, especially from the road-mobile sources. Afterward, reductions in air pollutant emissions were observed, especially from the road-mobile sources [3]. In the meantime, the environmental health management status of the workplaces may be poor because many workplaces are in small businesses and they are in the blind spot of legal regulations [4]. Environmental hazards in Seoul, such as traffic-related air pollution and occupational hazards in small workplaces, are often located close to residential areas. Workers in small businesses and residents in those areas may be more vulnerable.

According to the Environmental Health Act, the government shall prevent receptors including humans and environments from being damaged by environmental hazards and promote environmental health [5]. This Act also advocates providing information to citizens and guaranteeing the right to know. Under the Act, the Ministry of Environment in Republic of Korea established in 2020 a Second Master Plan for Environmental Health (2021–2030) to understand issues for environmental health and improve population health related to environmental hazards [6]. However, distributions of environmental hazards differ substantially by area. Therefore, the policy effect of a unified environmental health plan on major environmental health issues at the national level will vary depending on the region. Therefore, a customized plan considering regional characteristics is needed. The Act was amended on 5 January 2021, and has been in effect since 6 July 2021. The major contents of the amendment of the Act are to develop and implement a Local Plan for Environmental Health (LPEH) and to put a special focus on vulnerable populations such as children, residents living in more polluted areas, and the elderly population to improve their environmental health issues preferentially. Implementing an LPEH in Seoul would strengthen the role of local governments in promoting environmental health. The amendment put a special focus on children who are more susceptible than any other age group. Under this amendment, each local government has to develop and implement a Local Plan for Environmental Health. It would strengthen the role of local governments in promoting environmental health.

In order to improve policies for environmental health, the SMG should make community-customized plans. Understanding the characteristics of environmental health in Seoul is a pre-requisite. Environmental health projects in Seoul are carried out by various departments such as the Citizens’ Health Bureau, Water Circulation Safety Bureau, Climate and Environment Headquarters, Seoul Waterworks Authority, Economy and Employment Planning Bureau, and Seoul Research Institute of Public Health and Environment. Even a single environmental medium or product may be managed by various departments. Therefore, the department in charge of environmental health needs to promote systematic and sustainable environmental health policies through interdepartmental collaboration with a holistic point of view. In the LPEH, a detailed action plan for improving environmental health should be developed. In order to derive detailed action plans and guide the political direction of LPEH, SMG should establish a preliminary plan that has the nature of the master plan. The SMG should identify essential components to establish the preliminary plan. The components should include strategies, goals, tasks, etc. The components should be identified based on current environmental health conditions, opinions, concerns, and needs of citizens. As such, the SMG should conduct surveys and data analyses. In this paper, we aimed to present the preliminary plan that we developed to establish LPEH in Seoul, and the process of establishing the plan. The LPEH including detailed action plans will be established in the year of 2022.

## 2. Materials and Methods

### 2.1. Study Area

Seoul is located in the west-central part of the Korean Peninsula and lies within a topographic basin. Seoul has an area of 605.23 km^2^, 0.6% of the area of the Republic of Korea (Figure 1), and its population is about 9.7 million in 2021, accounting for 17% of the Republic of Korea’s population. From 2011 to 2020, the proportion of the child population in Seoul decreased from 13.9% to 10.6%, and the proportion of the elderly population increased from 9.8% to 15.4% (Figure 2). To explain the status of environmental health in Seoul, related data were abstracted from officially published studies and official websites. These data were analyzed using descriptive statistical methods.

### 2.2. Status of Environmental Hazardous Factors by Environmental Media [7]

We obtained annual average of air pollution data collected at 25 stations from the Seoul Air Pollution Monitoring Network [8], annual average of water pollution data at 17 stations from the Water Environment Information System [9], soil pollution data at 319 monitoring sites from the Seoul Urban Plan Portal [10], indoor air quality data from inspection history [11], the number of civil complaints about noise and vibration from SMG Portal [12], and light pollution data from the Seoul management status [13]. We also obtained the management status about children’s activity-space data from the related research report of the Seoul Institute [7]. The distribution of nine hazardous chemicals (e.g., ammonia), the distribution of small business sites such as printing presses and nail salons [14], and the distribution of green environments such as urban forest, were obtained from the SMG’s internal data [10].

### 2.3. Vulnerable Population and Environmental Diseases

The distribution of children less than 15 years of age and the elderly population more than 65 years of age was identified using demographic data [15] for each district. Data for social and economically vulnerable populations (e.g., recipient of living subsidies) were based on Seoul welfare statistics, and data for environmental diseases (e.g., allergic rhinitis and atopic dermatitis) were based on medical treatment data [16] obtained from the Ministry of Health and Welfare and National Health Insurance Service.

### 2.4. Analysis of Driving Conditions for Execution of Environmental Health Policies

To identify the current status of environmental health-related policies in Seoul, we collected related materials such as regulations, plans, and reports on implementation results. In detail, we analyzed the city’s environmental health investigation data, including environmental safety diagnosis and environmental consulting data for daycare centers, toxic chemical substances investigation for living-related workplaces, and reports on residents’ health impact assessment, etc.

### 2.5. Citizen and Expert Awareness Survey

On behalf of the SMG, we conducted a survey of Seoul citizens regarding their thoughts, concerns, and opinions regarding environmental health in Seoul to identify what environmental hazardous factors and environmental diseases should be prioritized and managed. In the survey, we investigated the perceptions and the experiences of awareness regarding risk, environmental pollution, hazardous chemical substances, environmental diseases and information provision, management direction, and coping effort [17,18]. To investigate the perception of environmental health and the experience of environmental health risks, 1,000 citizens who reside in Seoul were selected using a proportional allocation method by gender and age for each district of Seoul. Sampling and online survey company were conducted by the professional survey company Macromill Embrain which has citizen online panel.

## 3. Results

### 3.1. Status of Environmental Hazardous Factors by Environmental Media

The SMG has been measuring major air pollutants since 1978, and the measured values are announced through the official government websites. At present, 519 monitoring stations are distributed on the Korean Peninsula. In Seoul, 25 stations measure PM10, PM2.5, NO2, SO2, CO, and O3. To analyze the trend of air pollution in Seoul, we abstracted the annual average of air pollutant concentrations in Seoul from the official government website [19]. Annual average ambient concentrations of NO_2_ for the period 2009 to 2019 except the year 2012 were higher than the national standards (0.03 ppm for NO_2_) but lower since 2018 [8]. Annual average ambient concentrations of PM2.5 were higher than the national standards (15 μg/m^3^ for PM2.5) for the period 2016 to 2019 (Figure 3). In addition, intensity–duration–frequency of high PM2.5 concentration (lasts over 90 μg/m^3^ in 2 h) showed an increasing trend [8]. Despite SMG’s continuous efforts for improving air quality, the ambient concentration of ozone increased through time (Figure 4) [8].

The total organic carbon concentration measured at the 17 national automatic water pollution monitoring network points in Seoul showed little change among the points, which were within the national standards during 2019. Total phosphorus decreased through time at all points since 2015 [9]. According to the 2019 soil pollution survey conducted at 319 monitoring sites, lead, zinc, total petroleum hydrocarbons, and copper levels at eight sites exceeded the national standards. Six of the eight sites were areas where traffic-related facilities are concentrated [10].

For indoor air quality management, the SMG is applying the same or more stringent standards than the national standards. The SMG inspected indoor air quality for multiuse facilities with standards of PM_10_, PM_25_, CO_2_, HCHO, total culturable bacteria, and CO. Although the proportion of facilities exceeding the indoor air quality standards decreased from 3.7% for 2015 to 2.4% for 2018, the proportion for 2019 increased slightly (4.8% of the surveyed facilities in 2019). This was because the strengthened regulations began to apply in 2019 [11]. For noise and vibration, related complaints steadily increased from 21,745 in the year 2011 to 60,386 in the year 2020, and it was found that the main source of noise was a construction site or road noise [12]. All areas in Seoul are designated as a light-environment management area. As such, Seoul’s light pollution prevention plan was established in 2015 and has been managed so far [13].

Activity spaces for children need to be managed with more strengthened standards than facilities for adults. Some of the activity spaces for children (e.g., day care center, playground; 10 of 166 the surveyed subjects in 2018) exceeded the environmental safety management standards for, e.g., total culturable bacteria and CO [7]. This suggests that more careful management is needed. In addition, it is necessary to strengthen the management of children’s activity space by reflecting the amendment of the Environmental Health Act that strengthened phthalate and heavy metal standards.

The amount of hazardous chemicals from manufacturers such as waste treatment plants and printing and copying, moved to specialized processing companies in 2018, was 2.6 times higher than in 2013, suggesting that risk management should be strengthened in those companies. Furthermore, the number of businesses managing hazardous chemicals was 34,604 in 2018 and such businesses tend to be located close to residential and commercial areas. Most have only limited environmental regulations because they are small-scale businesses. Hazardous chemicals in such businesses must be regulated [14]. Regarding the green environment of Seoul, the urban forest per capita is the lowest (4.38 m^2^) compared to other cities in the nation, which is far below the WHO recommendation standard (9.00 m^2^) [10].

### 3.2. Vulnerable Population and Environmental Diseases

The age-vulnerable population of Seoul (children (age < 15) and the elderly (age ≥ 65)) and the socially and economically vulnerable population (number/percent of beneficiaries of government social assistance) are distributed throughout the city. The proportion of children is high south of the Han River, and the percentage of elderly and socioeconomically vulnerable population in the district population is high north of the Han River. According to population projection conducted by the SMG, the proportion of the elderly is expected to increase from 16.2% in 2021 to 30.1% in 2040 [20]. Economic polarization is expected to intensify because the income of the elderly is insufficient to live.

Concerning environmental diseases, the number of patients with allergic rhinitis and associated medical expenses increased through time in Seoul (Figure 5). For atopic dermatitis, despite the number of patients decreasing, the total amount of medical expenses increased (Figure 6).

Given the increasing elderly population and medical expenses, more detailed and comprehensive investigations on the surrounding environment should be conducted. Such investigations should consider distributions of diseases and causative substances, and distributions of vulnerable populations.

### 3.3. Analysis of Driving Conditions

Various departments in Seoul are in charge of environmental health services, such as the Citizens’ Health Bureau, Climate and Environment Headquarters, Water Circulation Safety Bureau, Seoul Waterworks Authority, and Seoul Research Institute of Public Health and Environment. Each department plays its own roles, for example, in establishing and implementing plans, managing vulnerable populations, conducting survey/evaluation/management on environmental health issues, reducing hazardous substances, constructing infrastructure, managing information, and collaborating with other organizations and institutions.

The SMG has conducted various environmental health investigations. For daycare centers (550 sites from 2015 to 2017), the SMG conducted environmental safety diagnosis and environmental consulting. The proportion of sites exceeding the phthalates standard was 94.3% in 2015, 68.2% in 2016, and 25.7% in 2017. Investigations for toxic chemicals were conducted for 17 small business facilities (e.g., nail shop) near residential areas. The rate of the nail shops not equipped with ventilation facilities was 14.7% in 2017. The percentage of workers who had experienced side-effects caused by products in nail shops was 39.6%. The SMG investigated health impact in residential areas affected by aircraft noise in the southwest region and around resource recovery facilities. There was no significant health damage identified in residents living around the airport and resource recovery facilities.

Although the SMG manages environmental health in various departments, the research supporting environmental health policies is still needed. In addition, the human resources, equipment, and budget are insufficient to carry out the work related to environmental health in Seoul. To overcome these problems, in the short term, some research should be conducted in collaboration with other institutions such as the Republic of Korea government-designated research centers for environmental health. In the long term, other research tailored to the characteristics of Seoul should be initiated, such as establishing Seoul’s own research center for environmental health or adding additional resources to SMG’s existing research institutes. Given that many citizens in Seoul use chemical products, the SMG should consider management plans for household chemical products.

### 3.4. Citizen and Expert Awareness Survey

To identify room for improvement in environmental health management, we conducted an online survey of 1000 Seoul citizens and 50 experts. Prior cases related to environmental health surveys [17,18] on risk perception, environmental pollution and hazardous chemicals, environmental diseases, information provision, management direction, and response efforts were reviewed to design the survey questions. As a result of the survey, both citizens and experts perceived PM as the most harmful (72.8% of citizens and 78.0% of experts), when compared to environmental hormones, water quality, heavy metals, climate change, harmful substances for children’s products, and noise and vibration. PM was found to be the most common problem experienced, in many dimensions: awareness, familiarity, personal knowledge, and scientific knowledge. For perception about environmental health issues, citizens considered PM (80.9%) as the most important regarding fear and severity. Experts (80.0%) recognized climate change, including global warming, as more fearful and severe than water quality, heavy metals, harmful substances in children’s products, or noise and vibration.

Citizens and experts recognized microplastics as a new environmental hazard more highly than other factors such as nanomaterials, persistent organic pollutants, natural radiation, and artificial light (72.4% of citizens and 88.0% of experts). Less than 40% of the citizens were aware of each new environmental hazard factor except for microplastics. More than 60% of the experts were aware of each new environmental hazard factor. Regarding environmental diseases, the citizens perceived that atopic dermatitis (82.0% of citizens), allergic rhinitis (79.8% of citizens), and cancer (74.1% of citizens) should be managed first in SMG. In addition, 26% of the respondents answered that they had experienced an environmental disease; 73% had experienced diagnoses of allergic rhinitis. More than 60% of the experts and less than 34% of the citizens thought that substances suspected of environmental hazards cause environmental diseases. The survey investigated the awareness of the age group that should be managed initially by environmental health policy of SMG. As a result, 68.1% of the citizens and 92.0% of the experts answered that infants and young children should be given priority. Less than 20% of the citizens and more than 50% of the expert group were aware of Seoul’s current environmental health-related policies or programs. Only 18.4% of the citizens were aware of public websites related to environmental health, but 76.7% of the citizens answered that they would be willing to participate in environmental health-related education courses.

### 3.5. Basic Direction for Implementation of the Preliminary Plan for Environmental Health in Seoul

We set the vision of the preliminary plan for environmental health in Seoul as “a safe Seoul environment, healthy citizens” with the meaning of securing a safe environment for all Seoul citizens to enjoy healthy living (Figure 7). This vision contains the meaning of preventing and managing damage caused by environmental hazards in advance, and establishing an essential environmental health foundation to ensure a healthy life for the citizens of Seoul. The strategies of the plan were set as “active monitoring of environmental health issues”, “minimization of health damage and meticulous and systematic response”, and “building a foundation for environmental health”. “Active monitoring of environmental health issues” means performing preemptively the tasks necessary to prevent health damage caused by environmental harmful factors. “Minimization of health damage and meticulous and systematic response” was presented to improve environmental health management of the vulnerable population, and establish compensation procedures for health damage caused by hazardous environmental factors. “Establishment of Environmental Health Foundation” was prepared to strengthen cooperation among related institutions for environmental health-related policy development in relevant research and to establish a system for upgrading information utilization.

### 3.6. Tasks Essential to Establishment of the Preliminary Plan for Environmental Health in Seoul

Based on the basic direction of the preliminary plan for environmental health in Seoul, we derived eight tasks and 25 subtasks under the three strategies (Table 1). Of the 25 subtasks, 12 subtasks were selected as key tasks. The key tasks were selected by collecting opinions of experts regarding the task’s importance, urgency, and feasibility. The main contents of the three strategies, eight tasks, and 25 subtasks are as follows.

This study presented the detailed five-year tasks arranged by year in consideration of the establishment of the business base, the urgency, and the connection with the current work. Some subtasks were made to be linked with other subtasks. A plan and budget were derived on an annual basis to consider the detailed tasks to be preceded and the detailed tasks to be followed. In Strategy 1, the task investigation of exposure to harmful environmental factors, the establishment of a basis for health impact investigation in areas of concern, investigation of toxic substances, and prevention of environmental diseases should be initiated, followed by other subtasks. Strategy 2 should be planned to start with strengthening the environmental safety management of the children’s activity space and improving the manual to prepare for environmental pollution and health damage. In Strategy 3, tasks promoting environmental health support center cooperative projects, maintenance of environmental health data, reinforcement of information usability, and development of Seoul environmental health indicators should be used as cornerstones so that detailed tasks can be continued.

To establish the plan for environmental health in Seoul, the planning components derived from this study are necessary, such as vision, strategy, and tasks. In addition, not only policy and institutional support, but also collaboration with related organizations is essential for efficient and systematic establishment of the plan. After the SMG finalizes LPEH based on this plan, it should notify the relevant departments. Additionally, in consideration of this plan and the Ministry of Environment’s National Comprehensive Environmental Health Plan, each department should establish a yearly detailed implementation plan. 

The performance of the task outcomes in LPEH should be assessed annually, and an interim evaluation will be conducted in 2026 after five years have elapsed. The evaluation is entrusted to a specialized agency to secure professionalism and objectivity, and the appropriateness of performance and direction. Based on the evaluation results, the implementation items are checked for each subtask and reflected in establishing the revised plan or the revised LPEH. In addition, results of annual performance index monitoring can be reflected in environmental health policies and projects at any time.

## 4. Discussion

To derive the planning components for environmental health in Seoul, we investigated the current status of environmental health, related research, and existing policies and programs in Seoul. In addition, we conducted online surveys targeting Seoul citizens and experts about the perception of environmental health issues and status in Seoul. Although SMG introduced various political options and partially improved some environmental media, citizens and experts recognized the need to further improve environmental health in Seoul. As a result of those investigations, we derived the vision, goal, three strategies, and 25 tasks of the preliminary plan for environmental health in Seoul.

The Seoul Environmental Health Committee entered into the process for establishing the Plan. The Seoul Environmental Health Committee was established in 2018 based on the ordinances in Environmental Health and the Right to Know the Community. The committee members include experts on environmental health and can discuss environmental health-related investigations and research, and deliberate on related plans. The committee participated in discussions for the vision, strategy, and tasks. The vision, strategies, and tasks of the preliminary plan were revised and supplemented by reflecting the committee’s opinions.

For the national level, the Ministry of Environment established the Comprehensive Plan for Environmental Health in 2006, the modified plan in 2011 and the second plan in 2020 [6,21,22]. In the second plan, the vision is “Safety Environment, Healthy Society”. Under this vision, the national plan is composed of four strategies, twelve main tasks, and fifty subtasks. For the community level, in 2013, the “Seoul City Environmental Health Policy Roadmap (SCEHPR)” [23] was established to investigate the adverse effects of hazardous environmental factors on human health and the environment, reduce the effects, prevent environmental diseases, and compensate those afflicted by environmental hazards. The Roadmap was composed of 4 strategies, 16 main tasks, and 45 subtasks. Researchers on SCEHPR selected twelve substances (7 major carcinogens, 5 environmental hormones) for priority management after consulting experts and collecting public opinions. In addition to the preliminary plans we have established, SMG will establish LPEH in Seoul by 2022 with consideration for understanding and connecting both the national Comprehensive Environmental Health Plan and the Roadmap for Environmental Health Policy in Seoul.

In Seoul, environmental health policies, especially related to air pollution, are already introduced and strengthened. In the early 2000s, Seoul experienced severe air pollution, mainly from PM and NO_2_. To improve air quality in Seoul and nearby areas, the SMG established a Master Plan for Atmospheric Environment Improvement in 2005 [24], and the Second Master Plan was established in 2015 [25,26]. Since the major air pollutants emission source in Seoul was a road-mobile source at that time, policies for controlling emissions from mobile sources were introduced in 2007. In the Second Master Plan, policies were introduced for reducing air pollutant emissions with consideration of changes in major emission sources and policies for the reduction in air pollution exposure, especially for the vulnerable population, such as children, elders, and outdoor workers. Therefore, these ongoing policies should be considered when establishing a detailed action plan under the LPEH.

In developing a set of action plans under the LPEH, the plans should consider the vulnerable population as a priory. Infants and children are susceptible to health impacts from exposure to hazardous environmental materials due to their immature body systems. In the Republic of Korea, many humidifier disinfectant-associated lung injury (HDLI) cases were reported. Attributable causing factors for developing HDLI are exposure to chemicals (PHMG (polyhexamethylene guanidine), CMIT (chloro-methylisothiazolinine)/MIT, etc.) contained in humidifier disinfectant (HD). From 2013 to 2017, 453 patients were identified with lung injuries developed due to chemicals in HD. Among them, 62% were aged less than eight years old and 31.7% were pregnant women [27]. Furthermore, exposure to HD during infancy is potentially associated with subsequent behavioral abnormality [28]. In the Republic of Korea, the total volume of chemicals distributed and the number of chemical species is constantly increasing. Considering the vulnerability and amounts of distributed hazardous chemicals, the action plans for chemicals, especially those used in products for a vulnerable population, should be given more attention.

The proportion of the elderly population is constantly increasing in Seoul (Figure 2). The elderly population is known to be vulnerable to the adverse effect of environmental hazards [29]. Among the elderly, those socially isolated and having low socioeconomic status are more vulnerable. Although the Environmental Health Act mainly targets children as a vulnerable population, the LPEH and detailed action plans should be developed with consideration for increases in the elderly population and its vulnerability. The elderly population spends much time indoors and may have chronic diseases such as cardiopulmonary diseases. Therefore, it is necessary to manage indoor air quality of frequently visited facilities such as welfare centers for the elderly. For example, it is necessary to establish a management plan by examining the actual exposure to polluted indoor air through an indoor air quality survey of welfare facilities for the elderly. In this study, measures for the elderly are presented in the preliminary plan as subtask 2-1-3 (Table 1).

Seoul’s LPEH should improve both environmental and health status within the metropolitan area. The SMG should regularly manage LPEH. The plan will stipulate that the SMG should evaluate the performance of the plan implementation at an annual and mid-term period. Since indicators for such assessment are not developed, further studies are needed to develop indicators. In this study, we presented the established preliminary plan and the establishment process. In this preliminary plan, we derived vision, strategies, main tasks, and subtasks to guide the future directions of environmental health policies to establish LPEH. The results of this work could contribute to establishing more effective and efficient policies in Seoul.

To develop the preliminary plan, we have reviewed various environmental health data. Nonetheless, there is a lack of data for many areas. In particular, although there are actual data, information related to environmental diseases and information on areas that exceed national standards, such as soil pollution, are not disclosed due to concerns over the adverse effects of information disclosure. In addition, data with higher resolution, such as census tracts, are needed to understand environmental health and to develop locally customized policies with consideration of local characteristics. To do this, we suggested gathering more detailed data with higher resolution in the third strategy of the preliminary plan. The accumulated data in the near future will contribute to following up local level environmental status and continuously modifying directions of environmental health policies.

## 5. Conclusions

This study was conducted to present blueprints and guidelines for the development of environmental health policies in Seoul, and to provide a basis for establishing LPEH in Seoul based on these findings. The blueprint presented in this study relates to the overall environmental health policy of Seoul during the next five years and corresponds to the vision, goal, and strategy. The contents of this study consisted of three steps: analysis of the current status of environmental health in Seoul, identification of implementation conditions, and deduction of tasks and implementation plans for each strategy. In addition, the scope of this study is to present the vision and goals of the SMG’s LPEH for five years until 2026, together with the annual focus and general tasks. To this end, analysis of related laws, systems, and policies, analysis of domestic and foreign current situations and cases, expert advice, and citizen perception surveys were conducted.

In this study, the vision of the preliminary plan for environmental health in Seoul was established in consideration of important keywords such as environmental diseases, citizens, and health. The vision, goals, strategies, and tasks were established based on the precautionary principle, the receptor-oriented principle, the principle of implementing environmental health justice, and the principle of guaranteeing participation and the right to know. Additionally, these were derived to ensure consistency with the contents of the Ministry of Environment’s Comprehensive Environment and Health Plan to facilitate cooperation with the Ministry of Environment and other local governments.

We anticipate that the preliminary plan for environmental health in Seoul will be beneficial for residents in Seoul, by protecting the health of citizens from the threat of hazardous environmental factors and by promoting environmental justice, thus satisfying citizens with the SMG’s environmental health policies.

## Figures and Tables

**Figure 1 ijerph-19-16611-f001:**
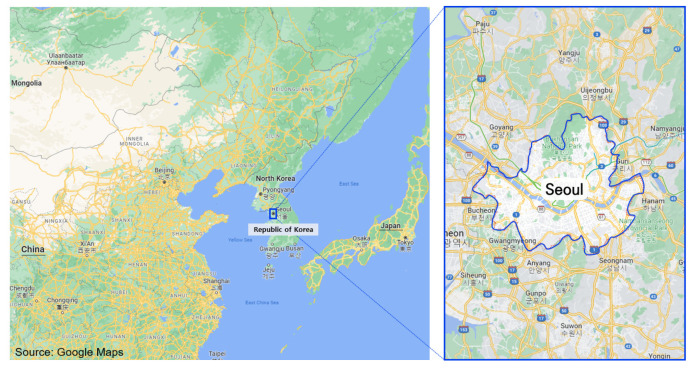
Study area.

**Figure 2 ijerph-19-16611-f002:**
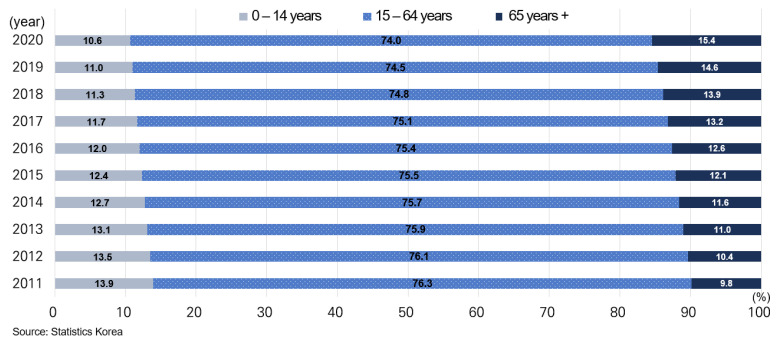
Population proportion by age group in Seoul.

**Figure 3 ijerph-19-16611-f003:**
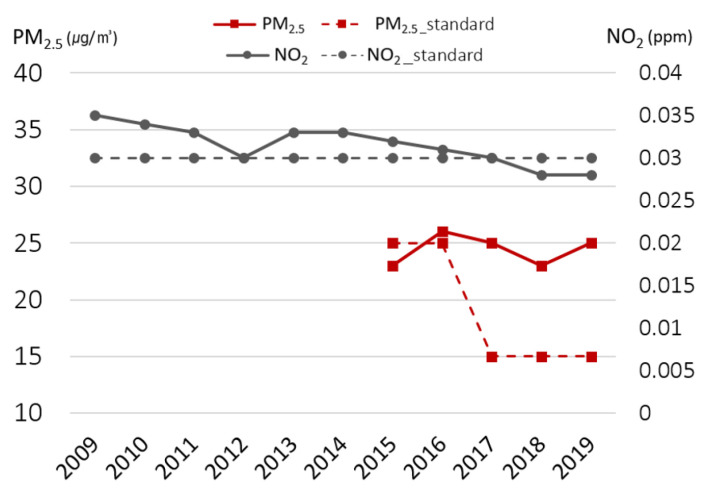
Concentrations of NO_2_ and PM_2.5_.

**Figure 4 ijerph-19-16611-f004:**
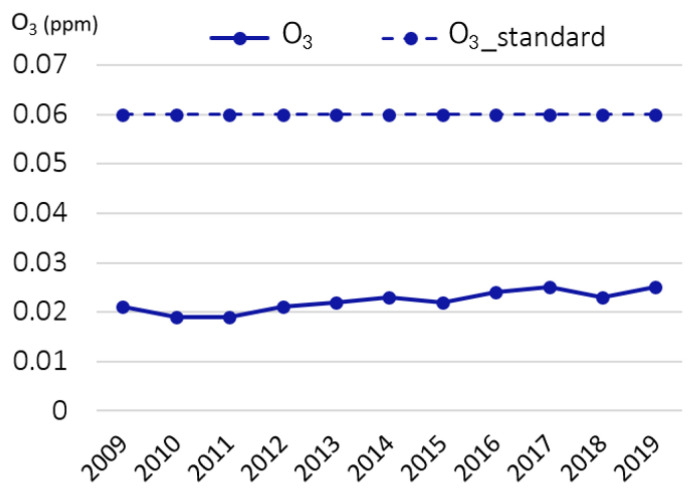
Concentration of O_3_.

**Figure 5 ijerph-19-16611-f005:**
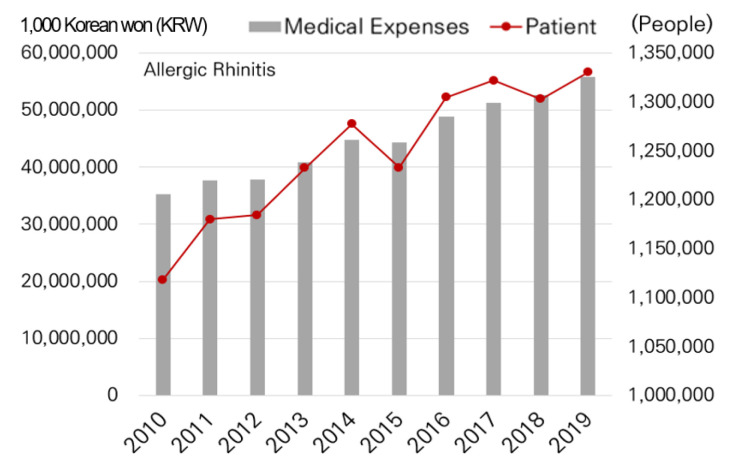
Allergic rhinitis patient and medical expenses.

**Figure 6 ijerph-19-16611-f006:**
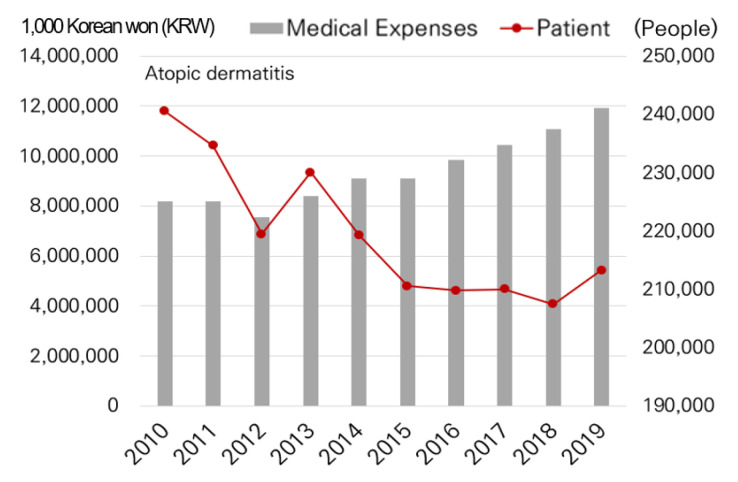
Atopic dermatitis patients and medical expenses.

**Figure 7 ijerph-19-16611-f007:**
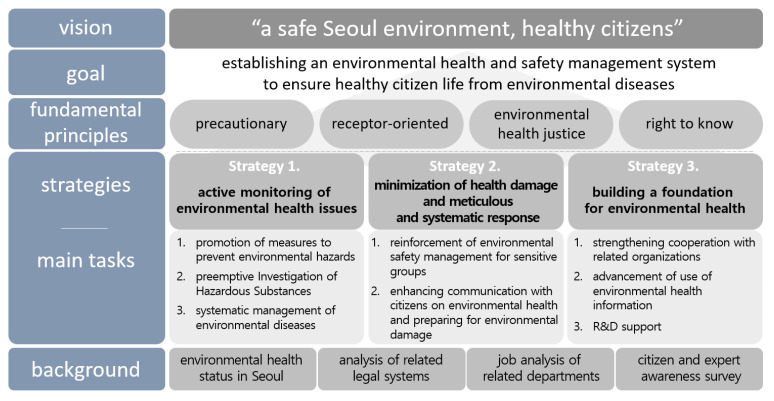
The basic direction of implementing the preliminary plan for environmental health in Seoul.

**Table 1 ijerph-19-16611-t001:** The importance, urgency, and feasibility of subtasks with experts’ evaluation.

Strategy	Subtasks	Importance ^(1)^	Urgency ^(2)^	Feasibility ^(3)^
1	1-1. Promotion of measures to prevent environmental hazards			
	1-1-1. Priority management on environmental harmful factors exposure survey *	4.2	4.3	4.0
	1-1-2. Investigation of health effects in areas of concern for environmental health vulnerability; build the foundation *	3.8	3.6	3.6
	1-1-3. Monitoring and management of hazardous factors in living environment *	3.9	3.9	3.8
	1-1-4. One-health monitoring on climate change	3.4	3.3	3.1
	1-1-5. Discovering new environmental harmful factors *	3.8	3.6	3.6
	1-2. Preemptive Investigation of Hazardous Substances			
	1-2-1. Investigating the actual condition of toxic substances and laying the foundation for management	3.5	3.4	3.5
	1-2-2. Measures for evaluating and monitoring hazardous substances in food preparation	3.5	3.4	3.6
	1-2-3. Establishment of foundation for safety management of household chemical products *	4.1	4.0	4.1
	1-2-4. Reinforcement of persistent pollutant emission management	3.6	3.3	3.1
	1-2-5. Establishment of the basis for managing antibiotic resistance in the environment	3.3	2.9	2.8
	1-3. Systematic management of environmental diseases			
	1-3-1. Strengthening the prevention and management of environmental diseases	3.8	3.6	3.3
	1-3-2. Promotion of Seoul-type Environmental Health Basic Investigation *	3.7	3.7	3.8
	1-3-3. Promotion of environmental health birth cohort project and follow-up of health effects by life stage	2.9	2.6	3.0
2	2-1. Reinforcement of environmental safety management for sensitive groups			
	2-1-1. Reinforcement of environmental safety management for children’s activity spaces *	4.6	4.6	4.6
	2-1-2. Elementary school environmental health cooperation project *	4.5	4.1	4.3
	2-1-3. Establishment of environmental health measures for the elderly and the socially and economically vulnerable population *	4.4	4.2	4.1
	2-1-4. Response to health damage caused by abnormal weather	3.7	3.7	3.6
	2-2. Enhancing communication with citizens on environmental health and preparing for environmental damage			
	2-2-1. Improving citizen communication on environmental damage and health impact	3.4	3.3	3.2
	2-2-2. Maintenance of the manual to prepare for environmental health and health damage	3.8	3.5	3.3
3	3-1. Strengthening cooperation with related organizations			
	3-1-1. Promotion of an environmental health support center cooperation project *	3.8	3.5	3.9
	3-1-2. Collaboration with international organizations for environmental health	3.2	2.9	3.6
	3-2. Advancing the use of environmental health information			
	3-2-1. Maintenance of environmental health data and utilization of information enforce *	4.1	3.6	4.2
	3-2-2. Establishment of Seoul environmental health information system *	4.1	3.9	4.1
	3-3. R&D support			
	3-3-1. Development of Seoul-type environmental health indicators	3.7	3.6	3.4
	3-3-2. Internet of things and AI-based environmental health policy leverage technology	3.4	3.2	3.7

* The key tasks were derived based on the value of importance, urgency, and feasibility exceeding the average. Experts gave one to five points in the survey. ^(1)^ Importance is the relatively more significant of the tasks. ^(2)^ The urgency is that it should be implemented relatively earlier among other tasks. ^(3)^ The feasibility is that it is easier for the Seoul Metropolitan Government to implement the task compared to other tasks.

## Data Availability

Not applicable.

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
