# Peer review of "A Study on the Preliminary Plan for Environmental Health in Seoul, Korea"

_ijerph, 2022, doi:10.3390/ijerph192416611_

Round 1

Reviewer 1 Report

This paper describes preliminary plan for environmental health in Seoul. From this study, it is expected that Seoul Metropolitan Government (SMG) is able to protect citizens' health from threats of environmental hazards, improve environmental health conditions especially in susceptible populations such as infants, and promote environmental justice. The methodological approach seems effective and appropriate. The paper was well organized and the results are important for the field of environmental science and public health. There is no problem regarding English. Therefore, I feel this paper should be acceptable after minor revision in view of the following specific comments.

(1)   The number of literature in “Introduction” is very few (only 2 references). Some previous or comparative studies either through literature or carried out in authors' own one must be cited.

(2)   There are no errors for data presented in Figures such as Figures 1-2. Please show measurement errors (i.e., error bars), if possible.

Reviewer 2 Report

Dear authors

I have carefully evaluated your manuscript. I consider the theme of your manuscript - environmental health in urban ecosystems - very relevant, especially in this historical moment in which most human beings live in cities, and need urban environmental quality.

I am inclined to recommend its publication to the editor of the IJERPH, but the manuscript needs some improvement, which I list below.

>>> The expression "Environmental Health" appears in the title and on all pages of the manuscript. However, at no time do the authors define what environmental health is, i.e., how it is defined. That expression must be defined at the beginning of the Introduction. Authors should keep in mind that IJERPH readers come from different backgrounds (they are biologists, physicians, geographers, managers, sociologists, etc.). So, what is obvious to authors may not be obvious to readers.

>>> IJERPH allows 10 keywords, you've included 7 in the manuscript. Please include 3 more keywords. They make the article more visible on search engines like Google Scholar, and help readers decide to read it.

>>> You mention the expression environmental justice in the Abstract: "we expect that SMG is able to protect citizens' health from threats of environmental hazards, improve environmental health conditions especially in susceptible populations such as infants, and promote environmental justice." I absolutely agree with the importance of promoting environmental justice to increase environmental health in cities. I have been doing research on environmental justice in cities in Brazil, and I was very interested in how the Local Plan for Environmental Health for the Seoul Metropolitan Government could promote it. However, the expression "environmental justice" appears only once more, in the penultimate line of the manuscript (line 384). I ask the authors how the Plan will contribute to promoting environmental justice in Seoul, and I strongly recommend that this answer be incorporated into the manuscript, in a paragraph or two.

>>> The acronyms SMG and LPEH appear frequently throughout the manuscript, which can make it difficult to read. Please follow the pattern on page two on the other pages where the acronyms appear: as in lines 48 and 51, the expressions must be written in full - Local Plan for Environmental Health and Seoul Metropolitan Government - and below, along the page, you could use the acronyms.

>>> In lines 46 and 47 you state that: "The amendment put special focuses on children who are more susceptible than any other age groups." The fact is that seniors are also more susceptible to environmental diseases. This leads me to my next recommendation:

>>> In Materials and Methods, please include a paragraph called "Study Area" and describe the city of Seoul: how many people are there in Seoul?; what are the percentages of children and seniors (I imagine there are more seniors than children, as in Japanese cities, am I correct? If I am, it seems surprising to me that The Act prioritizes children but not seniors); in addition, include all socio-environmental data that help readers understand Seoul and its socio-economic and environmental context.

>>> In the Results (page 4) you correlate vehicle traffic with atmospheric and noise pollution. In fact, this is a problem in many cities around the world. What do you propose to solve it, or at least mitigate it? This is not clear to me. Does Seoul have a good and efficient public transport system? In Brazilian cities, for example, public transport systems are bad, which leads citizens to travel in private vehicles, which emit more pollutants.

>>> In line 169 you state that "According to population projection conducted by the SMG, the proportion of the elderly is expected to increase from 16.2% in 2021 to 30.1% in 2040." This accelerated aging is also happening in many cities in Brazil, which is worrying, I mean, how can cities in Korea prepare for the aging of their citizens? How to promote environmental health for them? I strongly recommend that authors answer these questions in the Discussion.

>>> In lines 158-160 the authors state that: "Regarding the green environment of Seoul, the urban forest per capita is the lowest (4.38 m²) than other cities in the nation, which is far below the WHO recommendation standard (9.00 m²)." Surprisingly, the authors do not discuss how to increase urban forest per capita in Seoul. Green areas are of enormous importance to human health: for example, according to several scientific articles, residents of well-wooded neighborhoods exercise more than those living in treeless neighborhoods. On the other hand, neighborhoods with lots of trees and green areas usually concentrate upper middle class families, while poor families live in neighborhoods devoid of vegetation. If this is the case in Seoul, authors should write in Discussion about promoting environmental justice by increasing green areas in their neighborhoods.

Finally, I would like to congratulate the authors on the quantity and quality of the data you gathered and analyzed for the development of a preliminary environmental health plan for Seoul. These are not only quantitative data, but also qualitative ones, and together they provide a very precise picture of what needs to be changed, in favor of urban environmental health.

Reviewer 3 Report

The authors of the manuscript undertook an ambitious task aiming at: the analysis of environmental health conditions of Seoul, identification of driving conditions for implementation of environmental health conditions, setting policy directions, and establishing preliminary plan for environmental health. The scope of planned research seems to be very developed and requiring set of different actions needed to fulfill all of the expectations assigned to this research.

 General and minor comments:

·         The introduction should be more comprehensive. The authors refer only to 2 scientific papers, whereas it is expected that the topic will be presented in a broader scale, in reference to other up to date studies focused on similar topic. The aim of the introduction should be also the identification of so called “gaps of knowledge” that the authors are planning to complete.

·         The objective of the study should be clearly stated in the end of the “Introduction”. Now it is not clear.

·         The “Materials and Methods” section is not well presented and in some parts is even confusing. The authors mentioned that they used data concerning air pollution, water pollution, soil pollution, indoor quality,  light pollution, and complaints about noise and vibration in the city, but it is not explained on what basis these factors were selected and what scientific methodology stands for it. The same may be also assigned to the choice of chemicals mentioned in lines 75-76. In the “Materials and methods” chapter, the authors also mentioned that they conducted a survey of Seoul citizens to find out what are they opinions about environmental health. Nevertheless, it is not explained how was this survey conducted and how did the authors choose the target group of 1,000 citizens. I am also not sure whether the number of 1,000 citizens invited to the survey is representative for the city of over 10 million inhabitants.

·         In the “Results” section, the authors refer only to a few pollution indicators. It is not explained why only PM2.5, NO2, or O3 were regarded for the air. What about the rest of the indicators that may describe the quality of air, and impact the status of environmental health? Why were they omitted?

·         The information about soil quality is also too laconic. The authors mentioned that some indicators exceeded the standards (what standards?) at eight sites (what sites?). How can these contaminants influence the environmental health?

·         There is a lack of statistical analysis for the data that the authors gathered.

·         It is not clear how were the values of “importance”, “urgency” and “feasibility” assigned in Table 1.

·         The “Discussion” should be amended, as only 3 scientific papers were included in this chapter for confrontation of the results obtained.

·         In the summary, the authors should highlight the novelty and limitations of performed study, and delineate directions for further research.

·         Line 23: please check the font. “such as infants” seems to be different than the rest of the text. The same applies to line 40: “residents”.

·         The Act mentioned in lines 45-46 should be listed in the “References”.

Round 2

Reviewer 3 Report

The authors suplemented the manuscript and made corrections in accordance with the guidelines of the reviews. The article is clearer and more readable. I estimate the scientific value as average. The article may be accepted for publication in its current form. Good luck!

Author Response

Thank you for your review and comments.

I will finish the submission of my manuscript well.

Kind regards.